# Current Mirror Improved Potentiostat (CMIPot) for a Three Electrode Electrochemical Cell

**DOI:** 10.3390/s24185897

**Published:** 2024-09-11

**Authors:** Alexandre Kennedy Pinto Souza, Carlos Augusto de Moraes Cruz, Élvio Carlos Dutra e Silva Júnior, Fagnaldo Braga Pontes

**Affiliations:** 1Senai Institute of Innovation in Microelectronics (ISI-ME), Manaus 69075-000, Brazil; elvio.dutra@am.senai.br (É.C.D.e.S.J.); fagnaldo.pontes@am.senai.br (F.B.P.); 2Department of Electronics and Computation, Federal University of Amazonas, Manaus 69077-000, Brazil; carlosamcruz@ufam.edu.br

**Keywords:** potentiostat CMOS, amperometric sensors, electrochemical sensor

## Abstract

This work presents a novel compact CMOS potentiostat-designed circuit for an electrochemical cell. The proposed topology functions as a circuit interface, controlling the polarization of voltage signals at the sensor electrodes and facilitating current measurement during the oxidation–reduction process of an analyzed solution. The potentiostat, designed for CMOS technology, comprises a two-stage amplifier, two current mirror blocks coupled to this amplifier, and a CMOS push–pull output stage. The electrochemical method of cyclic voltammetry is employed, operating within a voltage range of ±0.8 V and scan rates of 10 mV/s, 25 mV/s, 100 mV/s, and 250 mV/s. The circuit is capable of reading currents ranging from 10 µA to 500 µA. Experimental results were obtained using a potassium ferrocyanide *K*_3_[*Fe*(*CN*)_6_] redox solution with concentrations of 10, 15, and 20 mmol/L, and their corresponding voltammograms were evaluated. The experimental results from a discrete circuit demonstrate that the proposed potentiostat topology produces outcomes consistent with those of classical topologies presented in the literature and industrial equipment.

## 1. Introduction

Electrochemical sensors are widely used across various measurement systems, spanning diverse fields such as the food industry, environmental monitoring, and biotechnology [1,2,3,4,5]. They are also integral to research in proteomics and genomics [2]. These sensors exhibit high sensitivity and selectivity in detecting a wide range of chemical species and organic compounds, including oxygen, glucose, and toxic metals [1,2,3,4,5,6,7,8,9,10,11,12,13]. 

Electrochemical sensors are designed to respond to specific quantities of chemical compounds or analytes of interest, generating electrical signals in the form of voltages or currents [3,8]. The amplitude of these signals is directly proportional to the concentration of the analyte in the sample, but this relationship is only valid when the concentration of the analyte is within the linear range. This process is characterized by the transfer of charge in the ionic solution through oxidation and reduction reactions [12,14,15,16,17,18].

There is significant interest in developing portable devices that integrate sensors and potentiostat electronic systems into a single unit. These devices would possess the necessary attributes for electrochemical instrumentation systems, including compact size, low power consumption, affordability, and high precision [19,20]. Efficient and reliable operation of such portable devices requires highly integrated electronic systems with low power consumption [20,21,22].

Examples of portable devices include those developed for detecting heavy metals in natural waters using electrochemical sensors to identify and quantify elements such as lead, mercury, and cadmium [4,23,24,25,26]. These sensors are vital for environmental monitoring and prompt decision-making in contamination scenarios [4,5,6,7,8,9,10,11,12,13,14,15,16,17,18,19,20,21,22,23]. Additionally, they can be applied to soil classification, enabling the identification and quantification of soil compounds for various environmental monitoring applications [4,23,24]. Moreover, implantable microsystems allow for continuous monitoring of organic compounds in human blood, including oxygen, glucose, and cholesterol [25]. Implantable glucose sensors are commonly used in diabetes management, oxygen sensors for monitoring tissue oxygenation in respiratory patients, and cholesterol sensors for controlling cardiovascular diseases [25,26,27,28,29,30,31].

These technological advances are enabled by electronic systems utilizing complementary metal oxide semiconductor (CMOS) circuit architectures and topologies, which integrate numerous electronic components into a single semiconductor chip [32,33,34,35,36,37]. This integration allows for the miniaturization and cost reduction of devices and makes CMOS potentiostats ideal suited for field measurements and portable applications. These devices are particularly effective for applications requiring precise electrochemical analyses, such as environmental monitoring and point-of-care medical diagnostics [38,39,40,41,42,43,44].

This paper proposes the design of a potentiostat using CMOS technology, employing an electronic circuit topology for the control and reading of signals in electrochemical sensors. The Materials and Methods section provides a detailed account of the electrochemical system, including a comprehensive description of the proposed CMOS potentiostat topology and an in-depth explanation of its technical characteristics. This approach proves the viability of the design for high-precision sensor applications. The Chemical Materials and Reagents section offers a thorough overview of the chemical compounds used in the experiments, confirming the suitability of the reagents to guarantee the reproducibility of the results. The potassium ferricyanide measurements subsection details the electrochemical methodology employed in the tests with this standard analyte, demonstrating the effectiveness of the potentiostat under controlled conditions. In the Test Environment section, we describe the equipment and controlled conditions used during the experiments, ensuring the validity of the results obtained. Finally, in the Results and Discussion and Conclusions sections, we present the results obtained through graphs and calibration curves, as well as a comparative performance analysis between the devices tested, concluding that the CMIPot potentiostat developed offers equivalent performance to the Autolab PGSTAT302N and represents a significant advance for the field.

## 2. Materials and Methods

### 2.1. Current Mirror Improved Potentiostat (CMIPot)

An electrochemical system consists of an assembly that includes a sensor device, or electrochemical cell (ECC), responsible for detecting chemical species sensitized by reactions in chemical compounds and an electronic circuit that provides the ECC bias and signal readout [1,7]. In this work, the sensor device is a three-electrode electrochemical cell (ECC), and the electronic circuit performing the ECC bias and signal readout is a potentiostat. The basic circuit of a (CMOS) potentiostat includes an operational amplifier used to apply controlled potentials in a three-electrode electrochemical cell and measure the current generated by the redox process occurring within the cell.

The scheme presented in Figure 1 illustrates a basic electrochemical system diagram comprising a three-electrode ECC, a potentiostat circuit, and their fundamental connections. To facilitate the testing process, the Tektronix AFG3252C function generator, Tektronic Inc., Beaverton, OR, USA, was used to generate a triangular waveform as the input signal to the system. A DC power supply provided the necessary power and biasing for the circuit, while the Tektronix MDO4104B-6 mixed domain oscilloscope, Tektronics Inc., Beaverton, OR, USA, was employed for data acquisition.

The functional block diagram of the CMIPot circuit is presented in Figure 2, providing a more detailed understanding of its components. The CMIPot potentiostat consists of three amplifiers, each serving different functionalities, two complementary PMOS and NMOS current mirrors, and a load resistor for reading the output signal. It also includes a dummy model of a three-electrode electrochemical cell.

The schematic representation of the CMIPot potentiostat is shown in Figure 3. The schematic employs a potentiostat topology with a novel approach to current measurement, developed explicitly for use in electrochemical cells.

A system with two cascaded amplifiers was devised to control the potentials across the reference (RE) and working (WE) electrodes. The first stage consists of a differential pair with a PMOS active load (M_1_–M_5_), while the second stage is a push–pull stage (M_6_–M_7_). The differential pair compares the control (Vin) and feedback (RE) signals, amplifying the resulting difference. The amplified signal is then passed to the push–pull stage for further amplification before being delivered to the electrochemical cell. The amplification process continues until the input voltage (Vin) is approximately equal to the reference voltage (RE). The current measurement of the electrochemical cell is performed by a set of transistors that form the PMOS and NMOS current mirrors within the circuit, in combination with the amplification stage composed of transistors M16, M17, and Rout. 

The printed circuit board (PCB) design for the CMIPot was developed according to specific requirements by implementing the connections shown in the schematic in Figure 3, using the CD4007UBE integrated circuit from Texas Instruments Incorporated, Dallas, TX, USA, as a basis for the physical implementation of the circuit. Figure 4a shows the 2D view of the top and bottom layers of the PCB, while Figure 4b illustrates the 3D view of the top and bottom layers. These complementary views provide a comprehensive understanding of the layout and placement of components. Figure 4c displays the fabricated PCB, highlighting the precise positioning of the CD40007 component and the arrangement of the connection traces.

### 2.2. Reagentes and Chemicals

All solutions were prepared using in ultrapure water obtained from a water purification system (Osmose Smart INOX, VEXER, Curitiba, Brazil), unless stated otherwise. The supporting electrolyte used in the experiments was potassium chloride (KCI), sourced from Dinâmica Química Contemporânea LTDA (Product ID: P.10.0294.072.01.27, Indaiatuba, SP, BR), and prepared at a concentration of 3M KCl at room temperature. The analyte used in the experiments was potassium ferricyanide, purchased from Lab-Synth (Diadema, SP, Brazil), and dissolved to create solutions with concentrations of 10 mmol/L, 15 mmol/L, and 20 mmol/L. Alumina polishes were obtained for electrode preparation.

### 2.3. Potassium Ferricyanide Measurements

To validate the CMIPot for electrochemical measurements, three potassium ferricyanide solutions with concentrations of 10 mmol, 15 mmol, and 20 mmol were prepared in a KCl-supporting electrolyte solution (pH 7.4) at room temperature. The device’s response was compared to that of a state-of-the-art commercial potentiostat (Autolab PGSTAT302N, Metrohm Autolab, Utrecht, The Netherlands) using cyclic voltammetry (CV) with the following parameters: starting and stopping potential of −0.8 V, a potential range of −0.8 V to +0.8 V, a step potential of 2.4 mV, with scan rates of 10 mV/s, 25 mV/s, 100 mV/s, and 250 mV/s. The electrochemical measurements were conducted using a gold (Au) working electrode (Metrohm: ID 6.1241.060), a silver/silver chloride (Ag/AgCl) reference electrode (Metrohm: ID 6.07726.100, and a platinum (Pt) counter electrode (Metrohm). A Faraday cage was employed during the experiments, and the solutions were maintained at room temperature.

All experiments were conducted using a 50 mL volume of potassium ferricyanide $K_{3}[Fe(CN{)}_{6}]$. This analyte is commonly used in a potentiostat evaluation due to its well-documented kinetic properties, which describe a reversible electrochemical behavior. Ferricyanide can be reduced to ferrocyanide, as illustrated by Equation (1), with the reverse reaction–oxidation of ferrocyanide to ferricyanide, as depicted in Equation (2).
(1)$$Fe{\left(CN\right)}_{6}^{3-}+e^{-}\to Fe{\left(CN\right)}_{6}^{4-}$$
(2)$$Fe{\left(CN\right)}_{6}^{4-}\to Fe{\left(CN\right)}_{6}^{3-}+e^{-}$$

### 2.4. Environmental Test and CMIPot and PGSTAT302N

Figure 5 illustrates the experimental setup for the electrochemical analyses conducted on both potentiostats. The test the proposed CMIPot potentiostat, a DC power supply (ICELL PS-500, Eletropeças Comercial Eletronica LTDA, Caxias do Sul, Brazil), a signal generator (Tektronix AFG3252C, Tektronic Inc., Beaverton, USA), a mixed-domain oscilloscope (Tektronix MDO4104B-6, Tektronic Inc., Beaverton, USA), and the three-electrode electrochemical cell with electrodes immersed in a potassium ferricyanide solution were utilized. The testing environment for the AUTOLAB PGSTAT302N commercial potentiostat includes a Faraday cage, the configuration of the three-electrode electrochemical cell with its associated connections, and the NOVA 2.1.6 Lab software, which is monitored using a notebook computer.

## 3. Results and Discussion

This section presents the results of the electrochemical analyses conducted as part of this project. A controlled experimental environment was meticulously designed, as shown in Figure 5, using the CMIPot PCB for a series of tests. The DC source was precisely tuned to a regulated voltage of ±6 V, providing optimal bias for the amplification and current mirror stages of the CMIPot circuit topology. The signal generator was configured for cyclic voltammetry, and the equipment was connected to the three-electrode electrochemical cell. To evaluate the system’s sensitivity, four different scan rates (10 mV/s, 25 mV/s, 100 mV/s, and 250 mV/s) were used with three concentrations (10, 15, and 20 mmol/L) of $K_{3}[Fe(CN{)}_{6}]$ solution.

A second testing environment was set up for data collection and comparative analysis, utilizing the AUTOLAB PGSTAT302N commercial potentiostat along with the necessary auxiliary equipment, as depicted in Figure 6. Cyclic voltammetry was applied with the same parameters described previously, allowing for performance comparison between the CMIPot and the commercial reference system.

The following graphs show the results of the electrochemical analyses using $K_{3}[Fe(CN{)}_{6}]$ solutions at concentrations of 10, 15, and 20 mmol/L, with potential scans at four distinct speeds: 250 mV/s, 100 mV/s, 25 mV/s, and 10 mV/s. The performance of the CMIPOT and the Autolab PGSTAT302N, was evaluated to provide a detailed comparison under identical experimental conditions.

In the first set of graphs (Figure 6a–c) applying cyclic voltammetry at a fixed scan rate of 250 mV/s and evaluating concentrations of 10, 15, and 20 mmol/L of K_3_[Fe(CN)_6_], revealed a proportional increase in current peaks with concentration. The calibration curves, obtained through linear regression, showed high coefficients of determination 0.999 for the CMIPot and 0.996 for the Autolab PGSTAT302N, Metrohm Autolab, Utrecht, Netherlands, indicating a robust linear response to concentration variations. Both devices exhibited similar current peaks and calibration curve slopes, demonstrating equivalent sensitivities.

In the second set of graphs, shown in Figure 7a–c, cyclic voltammetry was performed with a fixed scan rate of 100 mV/s for the same concentrations of 10, 15, and 20 mmol/L. The resulting cyclic voltammograms continued to exhibit a proportional increase in current peaks with concentration. The calibration curves for both devices at this scan rate displayed consistent and comparable coefficients of determination, with values of 0.9996 for the CMIPot and 0.9997 for the Autolab PGSTAT302N, confirming the accuracy of the measurements.

In the third set of graphs, presented in Figure 8a–c, the scan rate was fixed at 25 mV/s. The cyclic voltammograms for concentrations of 10, 15, and 20 mmol/L continued to show a proportional relationship between current peaks and concentration for both potentiostats. The calibration curves exhibited high coefficients of determination, with values of 0.996 for the CMIPot and 0.999 for the Autolab PGSTAT302N.

In the fourth and final set of graphs, displayed in Figure 9a–c, the scan rate was fixed at 10 mV/s. The analyses of 10, 15, and 20 mmol/L concentrations of K_3_(Fe(CN)_6_ showed that the CMIPOT and PGSTAT302N potentiostats maintained linearity in the current responses. The calibration curves, obtained through linear regression, exhibited coefficients of determination close to 1, with values of 0.995 for the CMIPot and 0.999 for the Autolab PGSTAT302N.

The comparative analysis of the CMIPOT and Autolab PGSTAT302N potentiostats, based on cyclic voltammetry data with K_3_[Fe(CN)_6_] solutions at concentrations of 10, 15, and 20 mmol/L, shows that both devices deliver equivalent and reliable performance under the tested conditions. At all tested scan rates (10 mV/s, 25 mV/s, 100 mV/s, and 250 mV/s), both the potentiostats demonstrated robust linear responses, with coefficients of determination near 1, indicating high precision and consistency. Both devices exhibited proportional current peaks relative to K_3_[Fe(CN)_6_] concentrations and showed comparable calibration curve slopes, signifying equivalent sensitivity. The results across different scan rates confirm the linearity and reproducibility of both devices, highlighting their capability for precise and consistent electrochemical measurements.

## 4. Conclusions

This study introduces a new compact CMIPot potentiostat circuit designed for electrochemical cells. The circuit functions as a control interface for the voltage polarization on sensor electrodes and for measuring current through the electrochemical cell. The proposed design features a two-stage amplifier, coupled NMOS and PMOS current mirrors, and a push–pull CMOS output stage, effectively capturing the current during oxidation–reduction process in analysis solutions.

The experiments, using the cyclic voltammetry with voltage ranges of ±0.8 V and scan rates of 10 mV/s, 25 mV/s, 100 mV/s, and 250 mV/s, demonstrated that the circuit accurately measures currents from 10 µA to 500 µA with high precision. Validation was achieved with a potassium ferricyanide (K_3_[Fe(CN)_6_]) redox solution at concentrations of 10, 15, and 20 mmol/L, enabling a detailed analysis of the resulting voltamograms.

The experimental results indicate that the proposed topology performs comparably to classical topologies in literature and industrial equipment. The circuit’s precision and consistency validate its effectiveness, showing that the new compact CMOS potentiostat is a viable alternative for electrochemical applications. The study confirms that the circuit delivers reliable and consistent measurements, meeting established technological standards. This lays a strong foundation for future research and advancements in CMOS potentiostats.

## Figures and Tables

**Figure 1 sensors-24-05897-f001:**
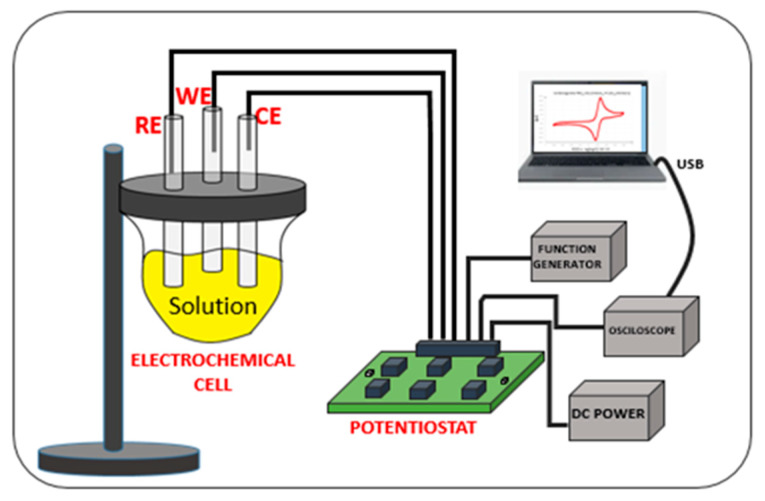
An electrochemical system containing the CMIPot PCB and a three-electrode electrochemical cell.

**Figure 2 sensors-24-05897-f002:**
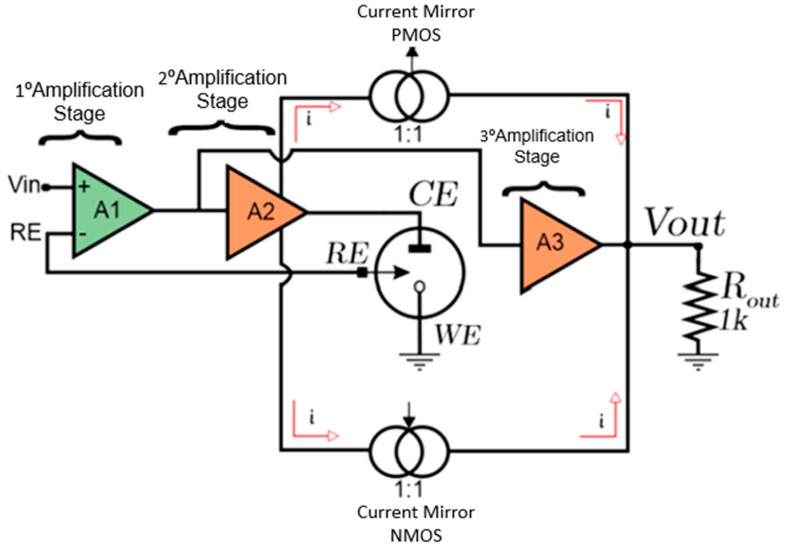
CMIPot Block Diagram.

**Figure 3 sensors-24-05897-f003:**
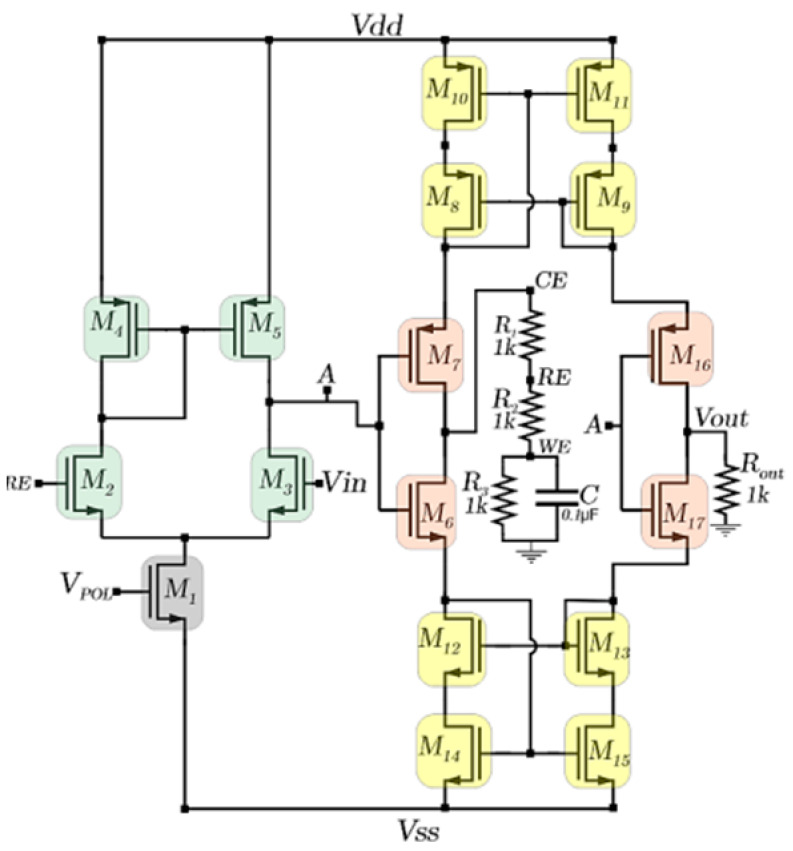
CMIPot schematic diagram.

**Figure 4 sensors-24-05897-f004:**
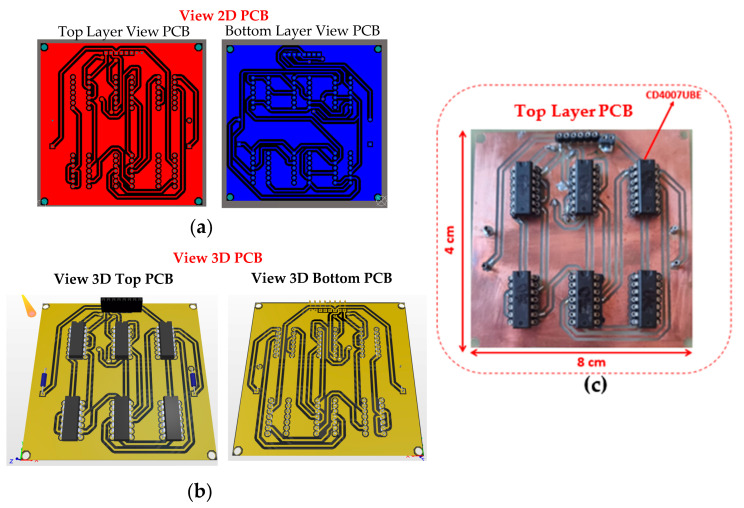
(**a**) 2D view PCB CMIPot; (**b**) 3D view PCB; (**c**) CMIPot fabricated PCB.

**Figure 5 sensors-24-05897-f005:**
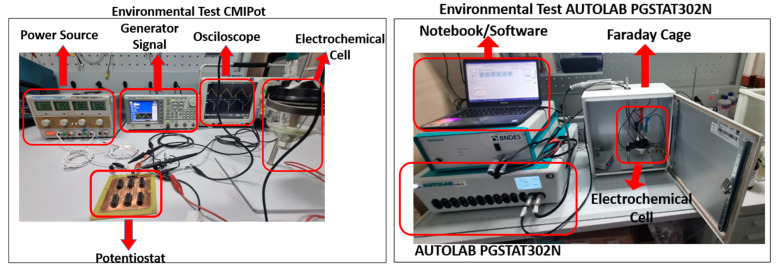
Environmental test and CMIPot and PGSTAT302N.

**Figure 6 sensors-24-05897-f006:**
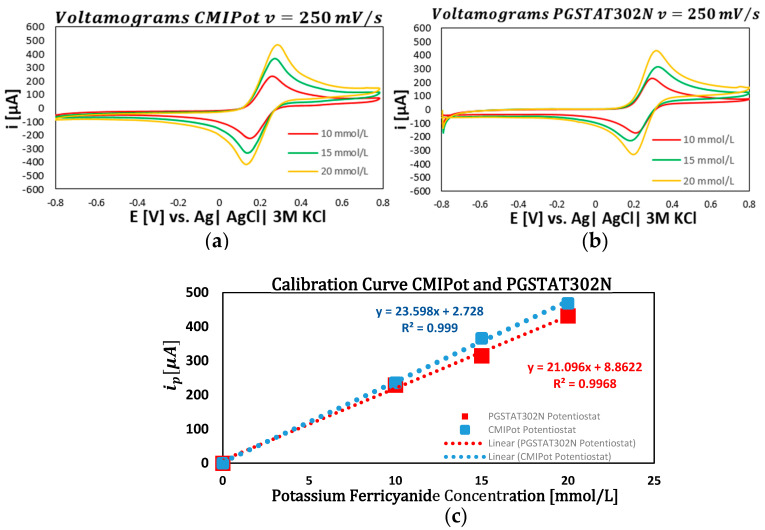
Voltamograms and calibration curve CMIPot and PGSTAT302N. (**a**) Voltammograms CMIPot with 250 mV/s; (**b**) voltammograms PGSTAT302N with 250 mV/s; (**c**) calibration curve comparisons CMIPot and PGSTA302N.

**Figure 7 sensors-24-05897-f007:**
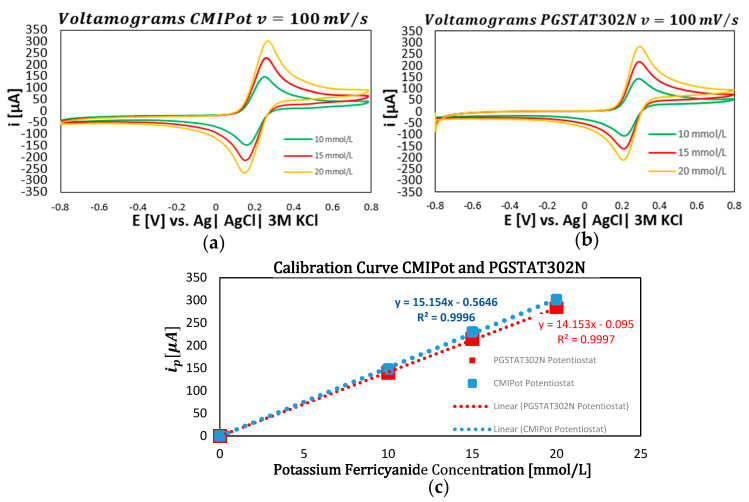
Voltamograms and calibration curve CMIPot and PGSTAT302N. (**a**) Voltammograms CMIPot with 100 mV/s; (**b**) voltammograms PGSTAT302N with 1000 mV/s; (**c**) calibration curve comparisons CMIPot and PGSTA302N.

**Figure 8 sensors-24-05897-f008:**
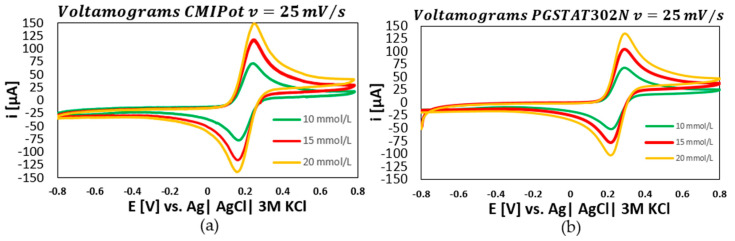
Voltamograms and calibration curve CMIPot and PGSTAT302N. (**a**) Voltammograms CMIPot with 25 mV/s; (**b**) voltammograms PGSTAT302N with 25 mV/; (**c**) calibration curve comparisons CMIPot and PGSTA302N.

**Figure 9 sensors-24-05897-f009:**
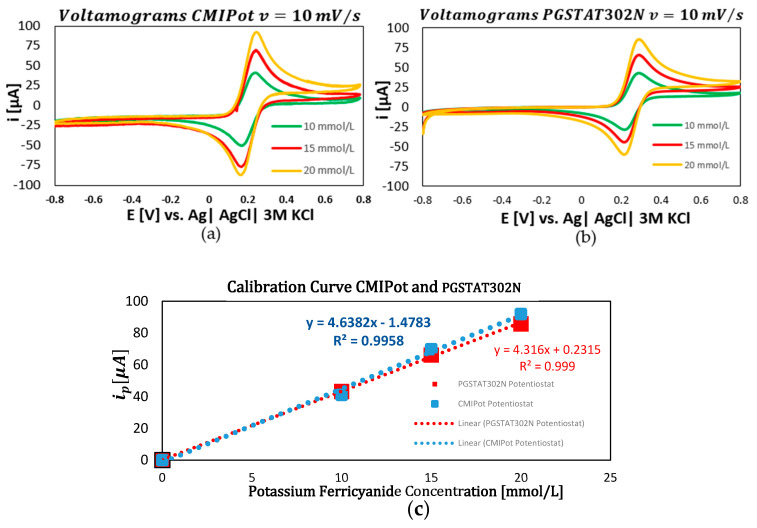
Voltamograms and calibration curve CMIPot and PGSTAT302N. (**a**) Voltammograms CMIPot with 10 mV/s; (**b**) voltammograms PGSTAT302N with 10 mV/s; (**c**) calibration curve comparisons CMIPot and PGSTA302N.

## Data Availability

The original contributions presented in the study are included in the article. Further questions can be addressed to the corresponding author.

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
