# Peer review of "Current Mirror Improved Potentiostat (CMIPot) for a Three Electrode Electrochemical Cell"

_sensors, 2024, doi:10.3390/s24185897_

Round 1
Reviewer 1 Report
Comments and Suggestions for Authors
The manuscript presents the development of an CMOS integrated potentiostat aiming for a portable device development. The concept of the idea is very interesting within the field of electrochemical instrumentation. Nevertheless, some aspects of the manuscript should be improved before considering publication:
1) Lines used for CV are too thick which harms proper visualization of the CV scans. Please reduce CV line width.
2) The CV is commonly performed using staircase CV, which would require the input of a step potential. What is the step potential (step in mV) in both experiments (CMOS and 302N)?
3) To which scan does the CV presented corresponds?
4) Why did the authors choose to use calibration plots as a correlation factor to the Autolab 302N? Why did the authors not choose thermodynamical related parameters such as peak-to-peak separation relation to Nernst Equation or Randles Sevcik square root relation between current and scan rate to assess the CMOS potentiostat correlation to 302N?
5) The depicted CMIPot CV presents a shift in E1/2 towards to more negative potentials compared to the same experiment performed with the 302N which occurs in all scan rates depicted. Please, provide an explanation regarding this matter.
6) It would be advised to perform electrochemical impedance spectroscopy to understand more of the performance of the proposed circuit as inductance cannot be observed in CV experiments.
Comments on the Quality of English LanguageThe English language must be reviewed throughout the entire manuscript as some sentences present poor construction and wrong verb tenses.
Reviewer 2 Report
Comments and Suggestions for Authors
In the paper “Current Mirror Improved Potentiostat (CMIPot) for Three Electrode Electrochemical Cell”, the authors proposed a new compact CMOS potentiostat circuit for electrochemical cell is proposed. In general, I think this paper can be published in the Journal of Sensors. However, a few things need to be taken care of before it can be published. Comments are given below:
1. In the abstract, you wrote, “…. With four concentrations chosen of 10, 15, and 20 mmol/L….” Instead of four different concentrations, you just tried three. I noticed this mistake happened multiple times throughout the whole manuscript.
2. The second paragraph of the Introduction section, you mentioned “The amplitude of these signals …………….and reduction reactions.” However, this is not rigorous. This is only correct if the concentration of analyte is within the liner range. Modify this sentence.
3. For the references, instead of giving all the references at the end of each paragraph together, you should give them at the end of each sentence. You should change the citations for the Introduction section (but not limited to this section).
4. The last paragraph of the Introduction section, you mentioned too many details about what you did for each section. But instead of these details, you should give more conclusive things. So, you should significantly change this whole paragraph.
5. In the Results and Discussion section, you compared the coefficient of determination between CMIPot and commercial PGSTAT302N. However, as I mentioned there is no meaning to compare this value between these two setups. As I said, you don’t know the concentrations (10, 15, 20 mmol/L) you tried here whether is within the liner range or not. So, there is no meaning to compare the coefficient of determination between CMIPot and commercial PGSTAT302N. Instead of this coefficient, you should make a detailed comparisons in like oxidation/reduction peak positions, cathodic /anodic peak currents, and others between these two setups. You can refer to this paper https://www.sciencedirect.com/science/article/pii/S0003267022010595 to know more about basic electrochemistry knowledge.
6. Fixed at one concentration, then try different scan rate (like 5, 10, 50, 100, 150, 200…. mV/s) to see the difference. For example, the relationship between square root of scan rate and the peak currents.
7. Besides the CV testing, you should also try the electrochemistry impedance spectroscopy (EIS) testing which is one very important in electrochemical sensors.
Comments on the Quality of English LanguageThe language is oaky.
Reviewer 3 Report
Comments and Suggestions for Authors
Réferee´s comments
To the paper entitled <<Current Mirror Improved Potentiostat (CMIPot) for Three Electrode Electrochemical Cell >> by Alexandre Kennedy Pinto Souza et al.
The paper is devoted to very interesting problem, which is important from the practical point of view. In this work, the design of a potentiostat in complementary metal-oxide semiconductor technology is proposed. The topology of electronic circuit for controlling and reading signals in electrochemical sensors has been developed.
Measuring systems, which involve electrochemical sensors, are used in medicine, pharmacology, food and beverage industry. They are also applied to ecological monitoring. Thus, the miniaturization of devices is required, since this approach makes them cheaper and available for all people. Particularly, it is possible to control a quality of air, water and food at home. These systems are also useful for the people, who have some problems with health. In order to reach miniaturization, a special organization of electric circuit is necessary. That is why a task of the work looks very attractive.
In order to control the potentials over the reference and working electrodes, two cascaded amplifiers were designed. The first stage consists of a differential pair with active load, while the second stage is involves push-pull. Potassium ferricyanide, which is common for the testing potentiostate, was used in this work as an electrolyte. The comparative analysis of the CMIPOT and Autolab PGSTAT302N potentiostats (commercialized), based on the data of cyclic voltammetry under different concentration of K₃[Fe(CN)₆] solutions, shows that both devices show equivalent, reliable and reproducible performance under all tested scan rates. Both potentiostats demonstrate current peaks, which are proportional to the solution concentration, the slopes of calibration curves are very close to each other. It means the proposed design of original potentiostate is adequate.
As for my opinion, the paper could be published after minor revision.
Please descript the CMOS abbreviation in the introduction section.
I would propose to omit Table 1, since it contains only one column of data. It is possible to give the information just in the text.
Regarding Figs. 7-9. I would propose to make lines in figures a-b much thinner, otherwise it is difficult to compare 2 potentiostates.
It is also recommended to merge Figs. 5 and 6 making Fig. 5a.
Conclusions
As I know, an Autolab PGSTAT302N potentiostate is rather large (comparable with a microwave stove). What is your opinion about the size of your potentiostate in a future? Can it be comparable with a cell phone?
Comments on the Quality of English LanguageEnglish must be slightly improved.
Round 2
Reviewer 1 Report
Comments and Suggestions for Authors
My previous suggestions and questions were properly addressed. Therefore, I recommend the manuscript for publication in its current form.
Author Response
Comments 1: My previous suggestions and questions were properly addressed. Therefore, I recommend the manuscript for publication in its current form.
Response 1: We would like to thank you for your time and effort in reviewing our manuscript and for your valuable suggestions for improvement and recommendations. Your contributions were fundamental in improving the clarity and quality of the paper. We are confident that the suggested adjustments have contributed significantly to strengthening the results presented and to a better understanding of the subject.
Reviewer 2 Report
Comments and Suggestions for Authors
Since the compact CMOS you built up is a replacement for potentiostat and all the work you have shown in the manuscript is its Direct current (DC) working performance, however, the Alternating current (AC) testing (like EIS) is completely missing. CV and EIS are two most used technologies, EIS is almost the most frequently used electrochemistry technology for some analytes that don’t have electroactive properties. Since you mentioned “This study introduces a new compact CMIPot potentiostat circuit designed for electrochemical cells.”, so I still think the circuit performance under the AC working condition is needed.
For the reference No.16, the format is not correct.
Author Response
Comments 1: Since the compact CMOS you built up is a replacement for potentiostat and all the work you have shown in the manuscript is its Direct current (DC) working performance, however, the Alternating current (AC) testing (like EIS) is completely missing. CV and EIS are two most used technologies, EIS is almost the most frequently used electrochemistry technology for some analytes that don’t have electroactive properties. Since you mentioned “This study introduces a new compact CMIPot potentiostat circuit designed for electrochemical cells.”, so I still think the circuit performance under the AC working condition is needed.
Response 1: We sincerely appreciate your valuable comment and the suggestion to include alternating current (AC) tests, such as Electrochemical Impedance Spectroscopy (EIS). We acknowledge that EIS is a technique frequently used for certain analytes that do not exhibit electroactive properties. However, as described in the initial objective of the article, the proposed potentiostat circuit topology is designed to measure the current in an electrochemical cell during the redox (oxidation-reduction) process.
Our objective was to develop and validate the circuit's performance under cyclic voltammetry (CV) conditions, which is an equally essential technique for the electrochemical characterization of systems with redox behavior.
That said, we recognize the importance of conducting tests under AC conditions and are considering incorporating this methodology in future studies to characterize electrochemical systems containing analytes without electroactive properties.
We hope this explanation clarifies the focus of the study and adequately addresses the questions raised.
We reiterate our thanks for the detailed review and for the valuable suggestions and recommendations for improving the article.
Comments 2: For the reference No.16, the format is not correct.
Response 2: We adjust according to the instructions.